# The reporting quality and spin of randomized controlled trials of endometriosis pain: Methodological study based on CONSORT extension on abstracts

**Hoda Shirafkan[1], David Moher[2], Parvaneh Mirabi[3]\***

**1** Social Determinants of Health Research Center, Health Research Institute, Babol University of Medical Sciences, Babol, Iran, **2** School of Epidemiology and Public Health, University of Ottawa, Ottawa, Canada, **3** Infertility and Reproductive Health Research Center, Health Research Institute, Babol University of Medical Sciences, Babol, Iran

\* parvaneh_mirabi@yahoo.com

**Data Availability Statement:** All relevant data are within the manuscript and its Supporting Information files.

## Abstract

### Objective

To assess the reporting quality of published RCT abstracts regarding patients with endometriosis pelvic pain and investigate the prevalence and characteristics of spin in these abstracts

### Methods

PubMed and Scopus were searched for RCT abstracts addressing endometriosis pelvic pain published from January 1st, 2010 to December 1st, 2023.The reporting quality of RCT abstracts was assessed using the CONSORT statement for abstracts. Additionally, spin was evaluated in the results and conclusions section of the abstracts, defined as the misleading reporting of study findings to emphasize the perceived benefits of an intervention or to confound readers from statistically non-significant results. Assessing factors affecting the reporting quality and spin existence, linear and logistic regression was used, respectively.

### Results

A total of 47 RCT abstracts were included. Out of 16 checklist items, only three items including objective, intervention and conclusions were sufficiently reported in the most abstracts (more than 95%), and none of the abstracts presented precise data as required by the CONSORT-A guidelines. In the reporting quality of material and method section, trial design, type of randomization, the generation of random allocation sequences, the allocation concealment and blinding were most items identified that were suboptimal. The total score for the quality varied between 5 and 15 (mean: 9.59, SD: 3.03, median: 9, IQR: 5). Word count (beta = 0.015, p-value = 0.005) and publishing in open-accessed journals (beta = 2.023, p-value = 0.023) were the significant factors that affecting the reporting quality. Evaluating spin within each included paper, we found that 18 (51.43%) papers had statistically non-

**Funding:** The authors received no specific funding for this work.

**Competing interests:** The authors have declared that no competing interests exist.

significant results. From these studies, 12 (66.66%) had spin in both results and conclusion sections. Furthermore, the spin intensity increased during 2010–2023 and 38.29% of abstracts had spin in both results and conclusion sections.

## Conclusion

Overall poor adherence to CONSORT-A was observed, with spin detected in several RCTs featuring non-significant primary endpoints in obstetrics and gynecology literature.

## Introduction

Randomized Controlled Trials (RCTs) are the best design to assess the efficacy *and* safety of therapeutic interventions in medicine; they are considered the foundation of evidence-based medicine [1].

The abstract of RCTs provides the reader with a summary account of the study objectives, methods, results, and conclusions. Often without access to full text articles, clinicians and others, only have access to read the abstract of a research report to help guide their clinical decision making. However, inappropriate study methodology with incomplete or inaccurate reporting impedes sufficient understanding of the clinical indications and restricts the reader's comprehension of the result's validity [2, 3].

These inferences are likely higher among physicians in low- and middle-income countries due to the unavailability of full-text publications. Similarly, patients and the public do not always have access to journal articles. Furthermore, determining whether a full-text publication is worth purchasing is crucial. These factors highlight the importance of reporting quality, clarity, and accuracy during the initial evaluation of RCT abstracts [4].

To improve the completeness and transparency of reports of RCT abstracts, the Consolidation of the Standards of Reporting Trials for abstracts (CONSORT-A), was developed in 2008 [5]. It provides the basic information to properly evaluate the validity of trials and the applicability and clinical relevance of trial findings. CONSORT-A includes 16 checklist items describing the spectrum of the research study, including information about the objectives, design, participants, interventions and their effect on primary efficacy outcomes and harms, conclusions, registration name and number, and source of funding [6].

Endometriosis is a painful, chronic inflammatory disorder. More than 60% of women diagnosed with endometriosis have dysmenorrhea or chronic pelvic pain. RCTs of endometriosis pain have not been critically evaluated [7]. To enhance the quality of reporting for RCTs in the endometriosis and gynecology literature, and in all intervention studies, researchers are motivated to publish their reports in clear, transparent, and unambiguous language. They must provide a precise and thorough description of participants, who were recruited, excluded, lost to follow up or did not complete the study. Because failing to declare essential data may influence the interpretation of results [8].

Furthermore, interpretation bias, often referred to as spin, is defined as misleading reporting of findings of a scientific study to highlight that the intervention is beneficial, despite the ignorable differences of the primary endpoint, or to confound the reader from statistically non-significant results. The tool for spin estimation in RCTs identifies reporting practices that constitute an intentional or unintentional effort to spin the results [9].

Although multiple RCTs have been published on the treatment of endometriosis pelvic pain, the adherence to the CONSORT guidelines and the prevalence of spin has not been

evaluated. Due to the prevalence of pain among endometriosis patients and the emphasis of guideline on conservative drug therapy instead of surgical interventions [10] as well as based on the above two statements the present study was designed to critically i) Assess the reporting quality of recently published RCT abstracts in endometriosis pelvic pain; (ii) Identify factors associated with reporting quality; (iii) Investigate the prevalence and characteristics of spin in these abstracts.

## Methods

In this study, we searched all abstracts published in PubMed and Scopus databases. Available full-text articles that published from January 1st, 2010 to December 1st, 2023 were chosen. We chose this time horizon to provide researchers with enough time to implement CONSORT for abstracts since its publication in 2008.

### Inclusion criteria

1. Study design:
   We included abstracts of primary, parallel or crossover of two-arm RCTs. We had no language restriction.

2. Participants:
   Adult patients (18–45 years old) with endometriosis associated pain. We include studies if the endometriosis were diagnosed with laparoscopy or surgery.

3. Intervention:
   We consider any interventions that affect the endometriosis associated pelvic pain.

4. Outcome:
   Studies with the primary or secondary outcome of endometriosis associated pelvic pain.

### Extra inclusion criteria for spin assessment

We included only studies with a statistical non-significant difference between group results for pain outcomes.

### Exclusion criteria

We excluded the Studies with the following criteria:

1. Animal studies

2. non-randomized trials, protocol studies, observational or cohort studies, review articles, interim or secondary data analyses of RCTs, or short surveys.

3. Letters, editorials, book chapters, thesis, conference papers, or news reports.

Two team members (H.Sh and PM) developed search strategy and selected the articles based on the following search strategy:

a. **PubMed**: ("Dysmenorrhea"[Title/Abstract] OR "painful menstruation"[Title/Abstract] OR "menstruation painful"[Title/Abstract] OR "pain menstrual"[Title/Abstract] OR ("Pain"[Title/Abstract] OR "pelvic pain"[Title/Abstract])) AND ("Endometriosis"[Title/Abstract] OR "Endometrioma"[Title/Abstract]) AND ("clinical trial"[Title/Abstract] OR "intervention study"[Title/Abstract]) AND 2010/01/01:2023/12/01[Date—Publication].

b. **Scopus:** (TITLE (dysmenorrhea) OR TITLE ("painful menstruation") OR TITLE ("menstruation painful") OR TITLE ("pain menstrual") OR TITLE (pain) OR TITLE ("pelvic pain")) AND (TITLE (endometriosis) OR TITLE (endometrioma)) AND (TITLE-ABS-KEY ("clinical trial") OR TITLE-ABS-KEY ("intervention study")) AND ((PUBYEAR > 2010 AND PUBYEAR < 2024) OR PUBDATETXT (**December 2023**))

In order to identify the articles, two researchers (PM and H.Sh) screened the retrieved articles. Then the duplicates were excluded. From each article, the required information (including number of authors, word count of abstracts, number of centers (single vs. multicenter), publication year, number of authors (less than 4, 4 to 7 and more than 7), and continent of origin (Asia, Europe, North America, South America and Australia) was retrieved. Furthermore, the journal name, indexing information of the journal (whether indexing in PubMed, Scopus, Embase, or DOAJ), being open access or not subject category (Specific Journal: Obstetrics & Gynecology vs. non-specific Journal), as well as the Journal metrics (impact factor (IF), IF Quartiles (Q1, Q2, Q3 and, Q4), cite score, H-index, frequency of citations in Scopus, and Field-Weighted Citation Impact (FWCI) were extracted (S1 File).

FWCI indicates the number of citations of an article compared to similar articles. Over three years, this is the ratio of a document's citations to the average number of citations for all similar documents. A FWCI greater than 1.00 indicates that the document's citations are higher than expected from the average citations of similar articles.

## Assessment of reporting quality

We used the extension of CONSORT statement for abstracts for assessing the reporting quality of included abstracts. If each of the items was reported adequately, we gave it a score of '1', and if the description was insufficient or incomplete, we gave it '0'. Therefore, the overall quality of reporting score (QoR; range from 0 to 16) was calculated for each included abstract. We also reported the 11 related sub-items of CONSORT-A [5].

## Evaluation of spin

Spin was defined as "using specified reporting procedure, of any motivation, in order to emphasize that the medication is useful, in spite of a statistically non-significant result for the primary outcome, or to confuse the reader from statistically non-significant results," as suggested by Boutron et al. [11].

We evaluated the spin respectively in the results and conclusions section of the abstracts. According to prior literature the spin types with the following items were assessed:

## Spins in the results section

- The focus of the results is on within-group statistically significant analysis;

- The concentration of the results is on the significance of secondary endpoints;

- Concentration of results relates only to the primary endpoint with statistical significance, although there are multiple primary endpoints.

## Spins in the conclusions section

- Claims of equivalence/non-inferiority/equivalence/similarity for statistically non-significant results

- Claims to receive benefits with a statistically non-significant result for the primary endpoint

- Focus only on statistically significant results (i.e., secondary endpoints, within-group analysis, subgroup analysis, analysis of a modified population)

- Identify primary endpoints that are not statistically significant but recommend the use of an experimental treatment

- If there are multiple primary outcomes, focus only on the statistically significant primary outcome

- If there are multiple primary outcome time points, focus only on statistically significant time points.

If the answer to any of the above questions is positive, that question is scored 1 point, otherwise it is scored 0. In each of the results and conclusion sections, we say that the section has a spin if at least one of the questions is answered affirmatively. Furthermore, in each of the results and conclusions sections, we considered the total points obtained to be equal to the spin intensity.

Spin was assessed by P.M. and H.Sh. independently. In the cases of disagreement, consensus was achieved by discussion.

The primary outcome must be clearly defined either in the abstract or in the full text. In the studies that the primary outcome was not clearly mentioned, we considered the declared outcome in the sample size calculations as the primary outcome. If not, we inferred a primary outcome based only on the expressed objectives of the study. In the case of secondary outcome or not identifying the primary outcome, we excluded the study from spin assessment.

## Outcomes

The main outcome of the study was the quality of the abstract reporting assessed using CONSORT-A. The overall reporting score is derived from the sum of items assessed in accordance with CONSORT-A guidelines. We also assessed the prevalence of spin in any section of the abstract as the secondary end point.

## Statistical analysis

Descriptive analysis was shown as frequency and percentages for categorical variables and mean, standard deviation (SD), median and, inter-quartile range (IQR) for continues variables. Assessing the associations between the study characteristics and the spin we used t-test, chi-square, Pearson correlation and, one-way ANOVA (with Bonferroni post hoc test). As the frequency of included articles was small or the distribution of the parameters was abnormal, we used the bootstrapping methods to assess the differences between groups. Bootstrapping is a with replacement sampling method. This method estimates the variations in a statistic by re-sample many times from observed data. We perform 1000 with replacement samples. Bias-corrected and accelerated (Bca) 95% confidence interval was computed.

In this study, we utilized a multivariable linear regression model to examine the factors influencing the Quality of Reporting (QoR) in the abstracts of Randomized Controlled Trials (RCTs) focusing on endometriosis pelvic pain. Understanding the determinants of reporting quality in scientific literature is essential for assessing the robustness of research findings. The independent variables included in our analysis were carefully selected to capture various aspects that could potentially impact the QoR of RCT abstracts. These variables encompassed characteristics such as the word count of the abstract, indexing in prominent databases like PubMed, Scopus, and ISI, Field-Weighted Citation Impact (FWCI), publication in a gynecological journal, open access status, Impact Factor (IF), Cite Score, and H-index.

Furthermore, to evaluate the strength and direction of the relationships between these independent variables and the QoR, we employed standardized coefficients, also known as Beta coefficients, in our multivariable linear regression model. Beta coefficients, or standardized coefficients, serve as a method to assess the magnitude and orientation of the association among variables that are denoted in disparate units or scales. The process of standardizing these coefficients normalizes the variables to a uniform scale, thereby simplifying the process of comparison.

Assessing the factors influencing the spin existence, we used logistic regression. We consider spin existence as dependent variable and the word count of the abstract, the indexing information of the journal (whether indexing in PubMed, Scopus, and ISI), designation as a gynecological journal (yes/no), publication in an open-access journal (yes/no), and the Journal metrics (impact factor (IF), cite score, FWCI, and H-index) as the independent factors. We used Statistical Package for Social Sciences (SPSS) version 23 to analyze the data. P-value less than 0.05 is considered as statistically significant.

## Results

Study Characteristics of the Selected Reports:

From 239 retrieved articles in databases, 47 clinical trials met the eligibility criteria and were included in the study. The study flow chart is shown in Fig 1.

Among these, most of articles were indexed in PubMed (45; 93.8%) and Scopus (45, 93.8%). Additionally, 43 (89.6%), 41 (85.4%) and 4 (8.3%) were indexed in EMBASE, ISI and ESCI, respectively. The article and the journal metrics for each study are shown in supplementary file (S1 File).

Assessing the characteristics of article abstracts, the average word count was 272.34 (±106.05) (min: 99, max: 734, median: 251, IQR: 78) and more than one-third (38.3%) of them had between 200 and 250 words. The mean of sample size was 227.59(±378.98) (min: 9, max: 1689, median: 83, IQR: 197). The mean Scopus citations was 41.11(±63.25) (min: 0, max: 321, median: 16). Taking into account the year of publication, document type, and disciplines the mean of FWCI is 3.24 (±4.05) (min: 0.14, max: 21.33, median: 1.70). Furthermore, we found that most articles described single center (n = 30, 63.8%) studies and were published in specific gynecological journals (n = 28, 59.6%). Almost half of the geographical area of the articles were in Europe and Asia (34.0% and 38.3%, respectively) (Table 1). Assessing the journal metrics, we found than the mean H-index of the journals was 167.20 (±198.08).

Evaluating the FWCI values according to the QoR (CONSORT values) shows a statistically significant correlation (r = 0.45, bootstrapping p-value: 0.003). As seen, the more the QoR the more is the FWCI.

Assessing the quality of the abstracts, we saw the total score for the quality varied between 5 and 15 (mean: 9.59, SD: 3.03, median: 9, IQR: 5). Furthermore, evaluating each item of CONSORT-A checklist, in titles of 26 (55.3%) papers, identified the study as randomized. In most of abstracts the eligibility criteria for participants was described. The detail of CONSORT items is shown in Table 2.

We compared the CONSORT-A total scores (QoR) for each indexing databases. Evaluating the quality of papers that indexed in Scopus, we observed a statistically difference between the scores of papers indexed in Scopus and papers that didn't index in Scopus (QoR = 9.77±3.03 vs. 7.00±1.73; bootstrapping p-value: 0.005; Bca 95%CI for mean difference (MD): 1.11, 5.04). Comparing the QoR between other indexing databases we did not see any significant differences (bootstrapping p-value>0.05). Furthermore, assessing the QoR in different IF quartiles we saw a statistically significant difference between them (QoR for Q1, Q2, Q3, and Q4

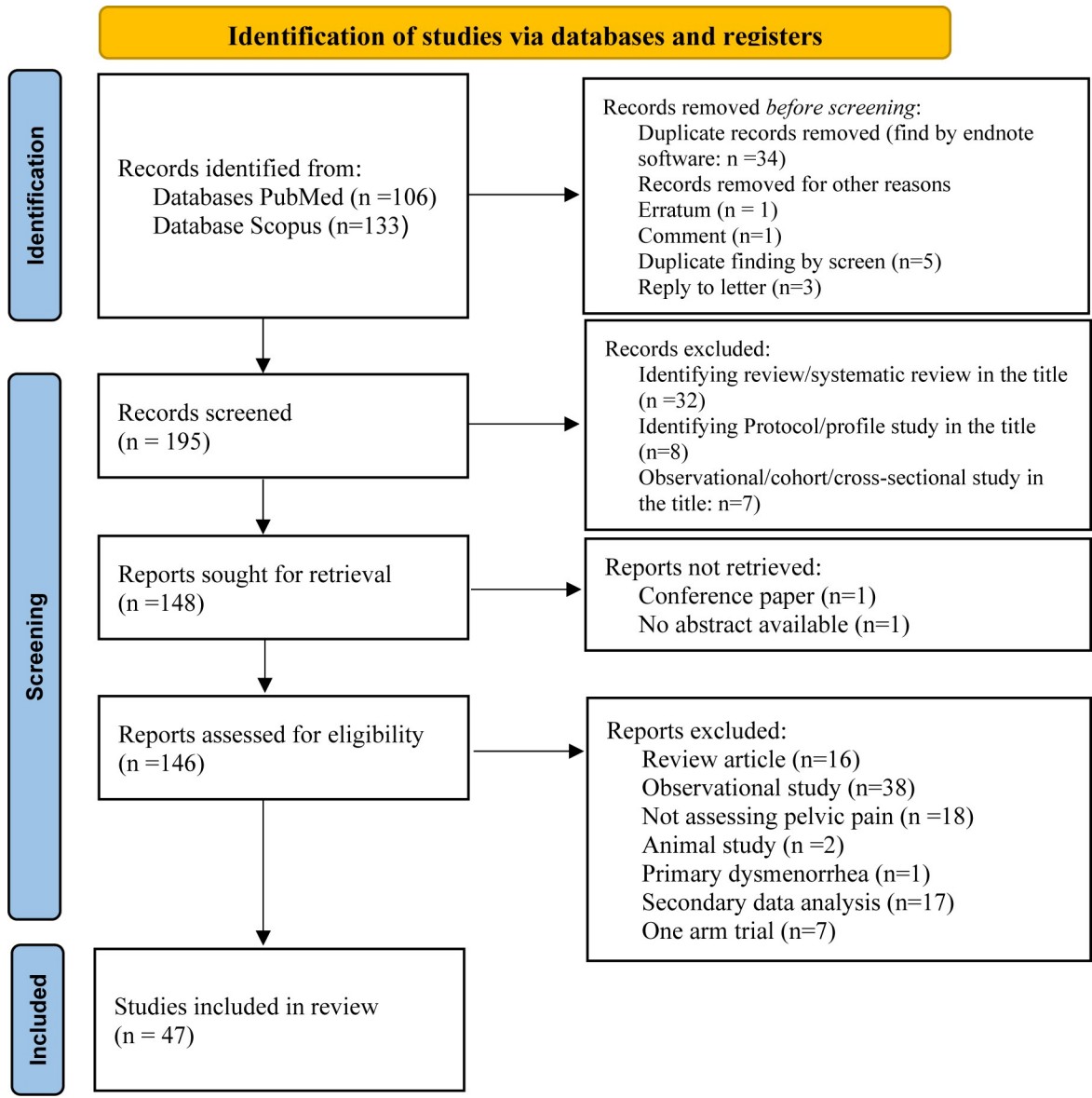

**Fig 1. Flow diagram of the study demonstrating the identification, screening, and inclusion stages.**

respectively were 11.75±2.53, 7.83±1.17, 8.36±2.73, and 6.67±0.58; bootstrapping p-value:<0.001) in which the QoR for articles that published in Q1 journals were statistically more than the other quartiles. Evaluating the QoR in different years and between IF quartiles, we saw that the QoR was increased for all quartiles except for Q4 (Fig 2). Also, assessing the QoR based on being open access or not, we saw that the QoR in open access journals were significantly more (QoR in open access journals: 10.52±2.78, QoR in not open access journals: 8.11±2.89, p-value:0.007).

A multivariable linear regression model was used to assess the impact of various independent variables on the Quality of Reporting (QoR) of Randomized Controlled Trial (RCT) abstracts. The independent variables included word count, indexing in databases (PubMed, Scopus, ISI), Field-Weighted Citation Impact (FWCI), publication in a gynecological journal,

**Table 1. Trial characteristics of the 47 endometriosis RCTs.**

| | Categories | Frequency (%) N = 47 |
|---|---|---|
| Year | 2010–2013 | 11(23.4%) |
| | 2014–2017 | 10(21.3%) |
| | 2018–2021 | 16(34.0%) |
| | 2022–2023 | 10(21.3%) |
| Centers | single center | 30(63.8%) |
| | multi-center | 17(36.2%) |
| Number of authors | <4 | 2(4.3%) |
| | 4–7 | 33(70.2%) |
| | >7 | 12(25.5%) |
| Word count | <200 | 5(10.6%) |
| | 200–250 | 18(38.3%) |
| | 251–300 | 12(25.5%) |
| | >300 | 12(25.5%) |
| Sample Size | <50 | 9(19.1%) |
| | 50–99 | 17(36.2%) |
| | ≥100 | 21(44.7%) |
| Type of journal | Specific Journal | 28(59.6%) |
| | non-specific Journal | 19(40.4%) |
| Open access | Yes | 29(61.7%) |
| | No | 18(38.3%) |
| continent of origin | Asia | 18(38.3%) |
| | Europe | 16(34.0%) |
| | North America | 6(12.8%) |
| | South America | 5(10.6%) |
| | Australia | 2(4.3%) |
| Impact Factor (6 articles didn't have IF) | <3 | 18(43.9%) |
| | 3–6 | 8(19.5%) |
| | >6 | 15(36.6%) |
| IF Quartile (7 articles didn't have IF Quartile) | Q1 | 20(50.0%) |
| | Q2 | 6(15.0%) |
| | Q3 | 11(27.5%) |
| | Q4 | 1(7.5%) |

open access status, Impact Factor (IF), Cite Score, and H-index. The goal was to determine significant predictors of reporting quality in RCT abstracts.

Our findings revealed valuable insights into the relationship between these independent variables and the QoR of abstracts. As shown in Table 3, we observed that an increase of one unit in the word count of the abstract was associated with a significant increase in the mean QoR (beta = 0.015, p-value = 0.005). Furthermore, our analysis indicated that abstracts published in open-access journals exhibited a mean QoR approximately 2.02 units higher than those published in other types of journals (beta = 2.023, p-value = 0.023).

## Spin evaluation

Evaluating spin within each included paper, we found that 21 (44.7%) papers had statistically non-significant results. From these studies, 15(71.4%) had spin in both results and conclusion sections. The mean total score of spin intensity was 2.67±1.46 than wasn't statistically different

**Table 2. The prevalence of reporting CONSORT-A checklist items.**

| Item | Description | N = 47 (%) |
|---|---|---|
| **Title** | Identification of the study as randomized | **26(55.3%)** |
| **Trial design** | Description of the trial design (e.g., parallel, cluster, non-inferiority)<br>a. no trial design description<br>b. description of trial design in method<br>c. description of trial design in title and method | **7(14.9%)**<br>**24(51.0%)**<br>**17(34.1%)** |
| **Methods** | | |
| **Participants** | Eligibility criteria for participants and the settings where the data were collected | **38(80.9%)** |
| | a. Eligibility criteria for participants | **30(63.8%)** |
| | b. the settings where the data were collected | **31(66.0%)** |
| **Interventions** | Interventions intended for each group | **45(95.7%)** |
| **Objective** | Specific objective or hypothesis | **45(95.7%)** |
| **Outcome** | Clearly defined primary outcome for this report | **38(80.9%)** |
| **Randomization** | How participants were allocated to interventions | **32(68.1%)** |
| | a. Assignment | **5(10.6%)** |
| | b. Sequence generation | **1(2.1%)** |
| | c. Allocation concealment | **1(2.1%)** |
| **Blinding (masking)** | Whether or not participants, care givers, and those assessing the outcomes were blinded to group assignment | **19(40.4%)** |
| **Results** | | |
| **Numbers randomized** | Number of participants randomized to each group | **18(38.3%)** |
| **Recruitment** | Trial status | **21(44.7%)** |
| **Number analyzed** | Number of participants analyzed in each group | **13(27.7%)** |
| **Outcome** | For the primary outcome, a result for each group and the estimated effect size and its precision | **11(23.4%)** |
| | a. primary outcome | **40(85.1%)** |
| | b. estimated effect size | **14(29.8%)** |
| | c. precision | **11(23.4%)** |
| **Harms** | Important adverse events or side effects | **13(27.7%)** |
| **Conclusions** | General interpretation of the results | **46(97.9%)** |
| **Trial registration** | Registration number and name of trial register | **21(44.7%)** |
| **Funding** | Source of funding | **12(25.5%)** |

between studies with and without funding (mean difference (MD) = 0.17, P-value = 0.82). The prevalence of spin among all included articles was 38.29 (18 from 47 articles at least had spin in one of results or conclusion section). The trend of changing the spin during 2010–2023 is presented in Fig 3. As seen in Fig 3, the spin intensity was increasing from 2010 to 2023. Assessing the spin in subgroups of impact factor, we saw that in articles that had been published in journals with IF more than 3, the trend of spin is decreasing from 2010 to 2023 while in the other IF categories, the spin is increasing.

No association was found with type of study area (obstetrics or gynecology) (p-value = 0.247), number of authors (p-value = 0.584), and being open access or not (p-value = 0.719) with spin.

To evaluate the factors influencing spin in the reporting of research findings, logistic regression analysis was conducted. The independent variables considered in the analysis included the word count of the abstract, indexing status in PubMed (yes/no), indexing status in Scopus (yes/no), indexing status in ISI (yes/no), FWCI, designation as a gynecological journal (yes/

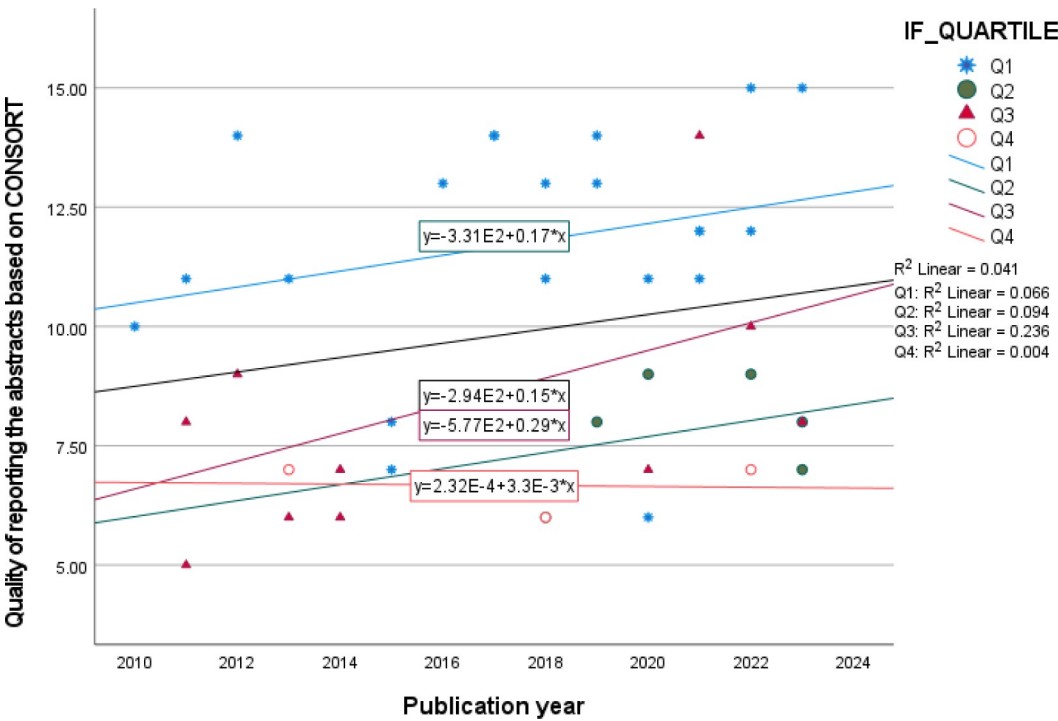

**Fig 2. Changing the quality of reporting the article abstracts according to different published years.**

no), publication in an open-access journal (yes/no), IF, Cite Score, and H-index. These variables were entered into the univariable linear regression model to assess their individual relationships with the presence of spin. However, the analysis did not reveal any statistically significant relationships between the independent variables and the presence of spin in the abstracts. Additionally, the model encountered convergence issues specifically for the variables related to indexing in PubMed, Scopus, and ISI, indicating potential limitations in the modeling approach for these factors (Table 4).

## Discussion

We conducted an evaluation of the reporting quality of RCT abstracts, analyzing the influencing factors and the prevalence and characteristics of spin in these abstracts. To the best of our

**Table 3. Factors affecting the quality of reporting the abstracts.**

| Parameters | Unstandardized Coefficients (β) | Standardized Coefficients | P-value |
|---|---|---|---|
| Word count | 0.015 | 0.530 | **0.005** |
| Impact factor | -0.245 | -2.81 | 0.231 |
| Cite score | 0.230 | 2.109 | 0.437 |
| Open access | 2.023 | 0.325 | **0.023** |
| H-index | 0.009 | 0.625 | 0.269 |
| gynecological journal | 0.911 | 0.146 | 0.389 |
| Indexing in ISI | -2.699 | -0.232 | 0.268 |
| Indexing in PubMed | 2.150 | 0.155 | 0.436 |
| Indexing in Scopus | 0.321 | 0.017 | 0.891 |
| FWCI | 0.137 | 0.186 | 0.503 |

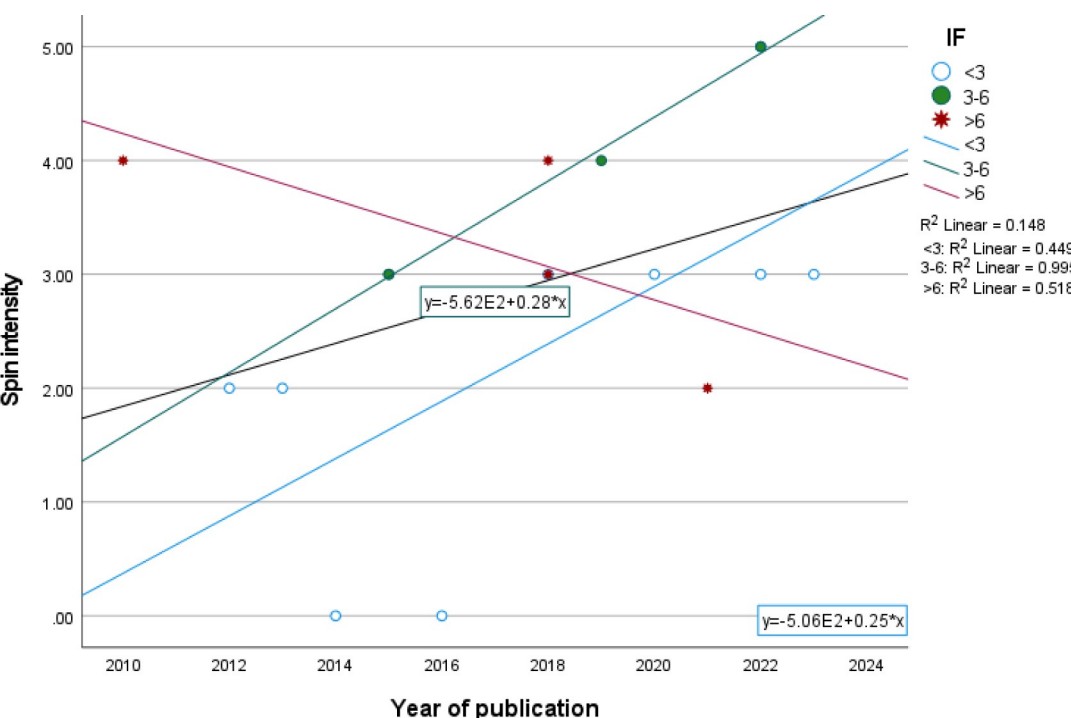

**Fig 3. Trend of changing the spin intensity during 2010–2023.**

knowledge, this is the first study examining the RCT abstract's adherence to the CONSORT statements regarding endometriosis-associated pelvic pain. The findings from this research are anticipated to offer valuable foundational insights for clinical decision-makers.

While CONSORT-A was designed to assist authors in improving the reporting of their RCT methods and results, our study revealed that the overall reporting quality of RCTs pertaining to endometriosis-related pelvic pain was found to be suboptimal. There is an apparent need to improve the quality of reporting of RCT abstracts, especially those related to key methodological domains.

Among the 16 checklist items outlined in CONSORT-A, only three items–namely, objectives, interventions, and conclusions–were adequately reported in the majority of abstracts (over 95%). However, none of the abstracts provided the required precise data. In the reporting quality of material and method section, trial design, type of randomization, the generation of random allocation sequences, the allocation concealment and blinding were most items identified that were suboptimal.

**Table 4. Factors affecting the spin existence in abstracts.**

| Parameters | Crude OR | 95% CI | P-value |
|---|---|---|---|
| Word count | 0.995 | 0.976–1.015 | 0.650 |
| Impact factor | 1.191 | 0.576–2.466 | 0.637 |
| Cite score | 0.992 | 0.655–1.502 | 0.969 |
| Open access | 0.625 | 0.048–8.201 | 0.720 |
| H-index | 1.009 | 0.989–1.029 | 0.395 |
| gynecological journal | 5.200 | 0.381–70.903 | 0.389 |
| FWCI | 0.137 | 0.186 | 0.503 |

Glasziou et al. recommended that all research funders and institutions need to change research rewards and regulations to align with better and more complete reporting, they should consider how best to participate in campaigns against wasteful studies. Research funders provide input of resources; however, outputs are managed by journal publishers with similar motivations as other publishers. Few employ a publication officer to promote research outputs, including attention to publication ethics and use of reporting guidelines [12].

A complete explanation of trial design can provide readers with an accurate research ideas and enable readers to assess the validity of the results [8]. However, in our analysis, we found that 50% of studies reported the type of trial design in the methods section, while 35.4% of studies reported it in both the title and methods sections.

It is particularly concerning that two of the most important points in the methods section (allocation concealment and blinding details) are almost completely neglected, as this information is the key data to certify the authenticity of results. We found only 1 (2.9%) study explained the methods of allocation concealment and 20 (41.7%) studies reported the details of blinding, which was similar to the results in the field of anesthesiology [13], acupuncture [14] and plastic surgery [15].

Furthermore, to identify the selection bias, it is important to indicate how participants were allocated and randomized to the intervention. This information is crucial for readers to assess whether the trial findings might be biased in favor of the new therapy [5]. Moreover, our study specifically focuses on RCTs that address the issue of endometriosis pelvic pain. Within the context of patient reporting outcomes, we recognize and emphasize the vital significance of implementing blinding protocols.

Of 48 studies, only one abstract explained the details of randomization, and although 30 (62.5%) studies mentioned randomized assignment, they did not provide adequate information about sequence generation, implementation procedures and allocation concealment, items that have not improved since the release of CONSORT-A. Additionally, taking into account the subjectivity and susceptibility of pain assessment in endometriosis, precise designs and better interventions should be implemented to avoid result exaggeration or overestimation.

Can et al. [16] stated that only 1.6% of RCT abstracts reported randomization methods, and similar reports were also demonstrated by other researchers in internal medicine fields [17, 18].

The primary outcome of a clinical trial serves as the foundation for sample size estimation, yet, to guard against data dredging, its pre-specification in the registered protocol and primary manuscript is essential. Data dredging refers to the exploratory practice of analyzing large datasets in search of any statistically significant associations, regardless of initial hypotheses. While the majority of reviewed RCT **abstracts (95.8%) outlined** study objectives, only **39 (81.3%)** explicitly stated the primary outcome, for the remaining abstracts, the primary outcome was extracted from the full text.

Since randomized controlled trials are included in medical guidelines and used by healthcare professionals, it is necessary to indicate the safety and effectiveness of the new therapy in the results section. On the other hand, as many specialists rely solely on abstracts of randomized controlled trials as the basis for clinical decision-making, it is important to mention adverse events in the abstracts [4]. Unfortunately, our analysis revealed that only 29.2% of the examined trials adequately reported significant harms or crucial adverse events in their abstracts. Furthermore, the analysis highlighted suboptimal adherence to two critical transparency components: trial registration and disclosure of funding sources.

Registration enhance reporting quality and makes trial information publicly available, thereby making authors responsible and helping to reduce publication bias and selective outcome reporting as it allows readers with limited access to full-text manuscripts to compare

abstract study findings with those defined a priori in studies [15]. However, our analysis identified that only 22 (45.8%) of the investigated RCT abstracts disclosed clinical trial registration information. While the impact of funding on research outcomes is well documented in existing literature, this critical information is typically absent from research summaries [5]. In this study, only 12 (25%) of the analyzed RCT abstracts disclosed their funding sources in this study. The lack of transparency regarding these critical details highlights a gap in reporting practices within the literature. It is important to consider the presence of confounding factors that may impact the reporting of this aspect of the manuscript. One such factor could be the word limit imposed by some journals on abstracts, which may hinder the comprehensive inclusion of all necessary information in RCT abstracts.

Our findings indicated a positive correlation between the word count of the abstract and QoR in RCTs addressing endometriosis pelvic pain. A more detailed and comprehensive abstract was associated with a notable increase in QoR, emphasizing the importance of thorough reporting in enhancing the quality of research summaries. Moreover, our analysis demonstrated that RCT abstracts published in open-access journals exhibited a significantly higher mean QoR compared to those published in other types of journals, with an approximate difference of 2.02 units. This underscores the potential impact of journal accessibility on reporting quality, underscoring the role of open access initiatives in fostering transparent and high-quality reporting in scientific research.

Interestingly, we also explored the influence of journal impact factor on reporting quality and found contrasting results compared to previous studies [4]. While higher impact factors were associated with better reporting quality in some research, our study did not find a significant impact of journal impact factor on reporting quality in the context of endometriosis pelvic pain RCT abstracts. Overall, our results provide valuable insights into the factors influencing the reporting quality of RCT abstracts in the context of endometriosis pelvic pain research. By identifying key predictors of reporting quality, such as word count and open access status, our findings contribute to the ongoing efforts to enhance transparency and accuracy in the reporting of clinical trials.

Furthermore, we observed that (15/21) of endometriosis pain RCTs with a non-significant primary outcome had evidence of spin in results and conclusion sections. Spin was more frequently observed in the conclusion sections of abstracts, and authors most commonly emphasized statistically significant secondary outcomes or misinterpreted a non-significant P-value as evidence of equivalence between intervention and control.

The implications of these results are manifest: Misrepresentation of study results misleads gynecologists into accepting interventions as beneficial despite non-significant primary outcomes, which can influence the decision and choice of a particular type of intervention.

The busy specialists and clinicians may not be able to identify and classify spin in the abstracts and may recommend rather unproven remedies.

Consistent with previous studies [19] no association was found with type of journal area (obstetrics or gynecology) and number of authors with onset of spin. Furthermore, no association has been observed between the presence of spins and funding sources.

There are few comparable reports in the field of obstetrics and gynecology. Chow et al. identified spin in 33% of abstracts from RCTs featuring non-significant primary endpoints across the five prominent obstetrics and gynecology journals, with a higher incidence of 46% in the conclusion sections [19]. These figures appear to be marginally lower than the findings in our study or those reported in the psychology literature [20], where spin was identified in 56% of the studies. Similarly, in the field of cardiology [21], spin was detected in 67% of the included texts.

As it mentioned in our study spin was found 66% in both results and conclusion sections of abstracts with non-significant primary outcomes published in 2010 and 2023. This is a notable finding that concerns the obstetrics and gynecology scientific community and it may be necessary to take safety precautions. To counteract the effects of spin and promote higher quality reporting in science, researchers, peer reviewers, clinicians and healthcare decision-makers should be made more aware of the presence of spin within RCTs. All journal editors should be aware and continue to monitor for spin to reduce the risk of reporting bias.

## Conclusion

This study adds to growing evidence that RCT abstracts in the gynecological literature may benefit from improvements, especially in smaller gynecologic subspecialties. Our results also suggest that spin is present in a number of RCTs with non-significant primary endpoints in the obstetrics and gynecology literature. The existence of such interpretive biases is of concern to the scientific community, as spins can influence decision-making in research, clinical, and healthcare systems. This highlights the need to advance strategies to counteract the effects of spin and promote better quality reporting in the scientific papers.

## Supporting information

**S1 Checklist. Human participants research checklist.**
(DOCX)

**S1 File. The article metrics and characteristics of journals [22–68].**
(DOCX)

**S1 Data.**
(SAV)

## Author Contributions

**Conceptualization:** Parvaneh Mirabi.

**Data curation:** Hoda Shirafkan, Parvaneh Mirabi.

**Formal analysis:** Hoda Shirafkan.

**Methodology:** David Moher, Parvaneh Mirabi.

**Software:** Hoda Shirafkan.

**Supervision:** David Moher.

**Validation:** Hoda Shirafkan.

**Writing – original draft:** Hoda Shirafkan, Parvaneh Mirabi.

**Writing – review & editing:** David Moher.

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
