## [Decision Letter · Decision Letter 0]

26 Dec 2023

PONE-D-23-21849The reporting quality and spin of randomized controlled trials of endometriosis pain: Methodological study based on CONSORT extension on abstractsPLOS ONE

Dear Dr. Mirabi,

Thank you for submitting your manuscript to PLOS ONE. After careful consideration, we feel that it has merit but does not fully meet PLOS ONE’s publication criteria as it currently stands. Therefore, we invite you to submit a revised version of the manuscript that addresses the points raised during the review process.

We look forward to receiving your revised manuscript.

Kind regards,

Ahmed Mohamed Maged, MD

Academic Editor

PLOS ONE

Journal Requirements:

4. We note that your Data Availability Statement is currently as follows: [All relevant data are within the manuscript and its Supporting Information files]

Reviewers' comments:

Reviewer's Responses to Questions

**Comments to the Author**

1. Is the manuscript technically sound, and do the data support the conclusions?

Reviewer #1: Yes

Reviewer #2: Partly

Reviewer #3: Yes

Reviewer #4: Yes

2. Has the statistical analysis been performed appropriately and rigorously? 

Reviewer #1: Yes

Reviewer #2: No

Reviewer #3: Yes

Reviewer #4: Yes

3. Have the authors made all data underlying the findings in their manuscript fully available?

Reviewer #1: Yes

Reviewer #2: Yes

Reviewer #3: Yes

Reviewer #4: Yes

4. Is the manuscript presented in an intelligible fashion and written in standard English?

Reviewer #1: Yes

Reviewer #2: No

Reviewer #3: Yes

Reviewer #4: Yes

5. Review Comments to the Author

Reviewer #1: Dear Authors

I read the presented manuscript with great interest.

Our team of researchers work on improving the methodological aspects of improving the reporting standards of RCTs and your manuscript adds value to the existing literature.

I suggest you the following changes to strengthen the quality of the study presented

1. It is better to include all the RCTs into spin evaluation to analyse the impact of significance of the results on the incidence of spin used by the authors

2. I suggest the authors to use the validated ORG-LOC tool which is developed to evaluate the level of confidence of the RCTs based on the spin present in their abstracts to better report the spin noted in the included studies

https://pubmed.ncbi.nlm.nih.gov/36310551/

3. Discussion has many paragraphs in single lines and i suggest reordering them based on content and clustering them together for ease of reading

4. I suggest you compare the CONSORT adherence and the LOC grade of the RCTs to obtaining some meaningful results to be conveyed to the readers

Reviewer #2: The authors analyzed the adherence to CONSORT for abstracts in 35 abstracts on RCT publications about endometriosis pelvic pain published between 2010 and 2021. They found that a mean of 8.54 from the 16 CONSORT-A was reported showing that there was poor overall adherence to CONSORT-A in these abstracts. In addition to that, they analyzed if studies with a non-significant primary endpoint misinterpreted this or if they reported secondary endpoints that were significant instead. They found that about 67% of these 18 abstracts had such a bias in the interpretation of results.

Even though the topic is important and the manuscript has some strengths, some points have been noticed while reading that should definitely be revised and adapted.

Flowchart:

The Flowchart that is shown in figure 1 should be improved. In the first 3 steps the reasons for exclusion of RCT-abstracts are not reported. This is necessary for the reader to understand the process of data collection and should be added. In addition to that the final number of abstracts for analysis is 35. This is correct for the analysis of adherence to CONSORT-A but as the authors write in their methods section they only included abstracts with non-significant primary endpoints (n = 18?). This should be added to the Flowchart to avoid confusion.

Abstract:

In the abstract and title the word spin is used for the sort of interpretation bias analyzed in the paper. This should be defined in the abstract as the meaning of spin might not be clear for every reader.

It is said that 35 abstracts are included but this wrong for the analysis of spin. Additionally the number of abstracts initially found should be added.

It should be defined if only the reporting of the CONSORT-A items is checked or if the quality/ completeness is also checked. If yes: How?

Introduction

It should be added that abstracts are not only the source of knowledge in countries with low income but also the decision maker if a fulltext publication is worth buying.

Methods

Please explain why you need information from fulltext articles if abstracts are of interest

Why do you chose a time of 2 years after publication of CONSORT-A? Why would 1 year not have been enough time?

Why are significant primary endpoints excluded from spin analysis? Is spin really absolutely impossible there?

open access yes/ no should be included as a characteristic and open access and non open access abstracts should be compared

Please define when a CONSORT-A item was reported adequately

Is the overall reporting score the sum of items? Please define it as it is the primary endpoint of the study. This should be changed in the section about the primary endpoint as this is used to identify the quality

To the 3 point in spins in the results section: how can this be checked when abstracts with multiple primary endpoints are excluded?

The last sentence in the “Spins in conclusions sections” should be part of the exclusion criteria

are all points according to spin of the same importance or is there a weighting possible/ necessary?

Statistical analysis:

Only the primary endpoint/comparison should have a hypothesis testing? → why are there so many tests planned?

In order to find associations and influencing factors on reporting quality a regression model (for example poisson regression) should be performed instead of the testing that is used. The authors should fit to regression models (one for CONSORT-A items and one for spin) so that a better (and adjusted) interpretation of the factors is possible. I ask the authors to add to these regression analyses.

Results

The adherence to CONSORT-A should be described in more detail as it is the primary endpoint of the study.

Please delete testing and add the results of regression modeling

Discussion

When you analyze factors that affect the quality then this should be done with regression models

General question: Did you check if the item was reported or was the quality and completeness of the reporting checked as well?

“none of the abstracts presented precise data as required” what do you mean with this statement?

“most items identified that were suboptimal” it was not reported at all or insufficiently

why are allocation concealment and blinding the most important points?

Please add the information on the medical field that was investigated in reference 17 and 18

Why was information on endpoints extracted from the full texts when abstracts were reviewed?

I could not find the results on IF that are mentioned in the discussion

The study registration is mentioned as a method to improve quality. But how is ensured that these are complete and correct?

Conclusion:

The study only shows the need for improvement of RCT abstracts but not of RCT reports. Please change the conclusion accordingly.

The own results and the interpretation should be discussed in more detail

The authors should revise the language to improve readability in some parts of the manuscript.

Reviewer #3: Dear Editor in chief

Thank you for the opportunity to review this manuscript. There are some specific comments:

Method:

- Please update the search. The search is for 2 years ago.

Discussion:

- Page 19: "Only one abstract explained the details of randomization, and although 20 (57.1%) studies mentioned randomized assignment, they did not provide adequate information about sequence generation"

Many journals have word limits and it is not possible to write details of randomization.

Reviewer #4: The study addresses one of the important areas of reporting of abstracts of clinical trials in endometriosis. The study is merthodologically sound and well presented, however, I have one major concern which is the period of study for article selection. The authors indicate that the articles published until Sept 2021 has been included. It is possible that the authors might have missed some crucial data on the trials published in the past 2 years. Hence I suggest the authors to update the search by including articles published until September 2023 and present the updated results.

6. PLOS authors have the option to publish the peer review history of their article (what does this mean?). If published, this will include your full peer review and any attached files.

Reviewer #1: **Yes: **Dr Sathish Muthu

Reviewer #2: No

Reviewer #3: **Yes: **Shabnam ShahAli

Reviewer #4: No

---

## [Author Response · Author response to Decision Letter 0]

14 Feb 2024

We highly appreciate the reviewers' and editors’ insightful and helpful comments on our manuscript entitled “The reporting quality and spin of randomized controlled trials of endometriosis pain: Methodological study based on CONSORT extension on abstracts”. We appreciate the opportunity to address the reviewer's comments. Extending our search to include articles published until December 2023 was indeed a challenging and time-consuming process. Despite the difficulty, we were able to identify and include 12 additional papers, which have been thoroughly analyzed and integrated into the revised manuscript. We believe that these updates have significantly enhanced the comprehensiveness and robustness of our findings., and we are confident that these revisions will strengthen the overall contribution of our study to the field.

Flowchart:

The Flowchart that is shown in figure 1 should be improved. In the first 3 steps the reasons for exclusion of RCT-abstracts are not reported. This is necessary for the reader to understand the process of data collection and should be added. In addition to that the final number of abstracts for analysis is 35. This is correct for the analysis of adherence to CONSORT-A but as the authors write in their methods section they only included abstracts with non-significant primary endpoints (n = 18?). This should be added to the Flowchart to avoid confusion. 

Response:

The flowchart now incorporates the rationales for excluding RCT abstracts. Additionally, the data collection procedure has been incorporated and emphasized. Out of the initially identified 47 eligible abstracts, all were ultimately included in the final analysis. However, during the spin assessment, it was observed that only 21 papers (in new search)exhibited non-significant primary endpoints. To enhance transparency, a dedicated step for spin analysis has been introduced to the flowchart.

Abstract:

● In the abstract and title the word spin is used for the sort of interpretation bias analyzed in the paper. This should be defined in the abstract as the meaning of spin might not be clear for every reader. 

Response: We added the definition of spin to the abstract and highlighted.

● It is said that 35 abstracts are included but this wrong for the analysis of spin. Additionally the number of abstracts initially found should be added. 

Response: We conducted a two-phase analysis of the included abstracts. Initially, we scrutinized the adherence to CONSORT guidelines in 47 eligible abstracts. Subsequently, all incorporated papers underwent an assessment for spin, revealing that only 21 papers featured non-significant primary endpoints. 

● It should be defined if only the reporting of the CONSORT-A items is checked or if the quality/ completeness is also checked. If yes: How?

Response: Our study's main aim was to evaluate the reporting quality of RCT abstracts, focusing on CONSORT-A criteria.

Introduction

● It should be added that abstracts are not only the source of knowledge in countries with low income but also the decision maker if a fulltext publication is worth buying. 

Response: In response to the reviewer's suggestions, the document was revised and highlighted.

Methods

● Please explain why you need information from fulltext articles if abstracts are of interest.

Response: To assess both primary and secondary outcomes during spin analysis, a thorough examination of the full-text articles is necessary. The primary outcome should be explicitly stated in either the abstract or the full text. In cases where the primary outcome is not clearly defined, we relied on the outcome declared in the sample size calculations as the primary measure. If such information was not available, we inferred the primary outcome solely from the stated objectives of the study.

● Why do you chose a time of 2 years after publication of CONSORT-A? Why would 1 year not have been enough time?

Response:This time frame allows ample opportunity for researchers to implement CONSORT for abstracts effectively. Additionally, it accounts for the typical duration encompassing peer review processes and the final publication in journals. This extended period ensures a comprehensive evaluation of the integration and adoption of CONSORT-A guidelines within the scholarly community.

● Why are significant primary endpoints excluded from spin analysis? Is spin really absolutely impossible there?

Response: As the definition of the spin, it may occurs when the between groups effect were non-significant. The exclusion of significant primary endpoints from the spin analysis was not due to the belief that spin is absolutely impossible in those cases. Rather, our decision was based on the methodological focus of our study, which specifically aimed to assess the reporting quality and spin of randomized controlled trials of endometriosis pain. As such, our analysis was centered on the reporting of outcomes and the potential influence of spin on the interpretation of trial results, rather than the presence or absence of statistically significant findings.Therefore, the articles that had significant primary endpoints excluded from spin analysis.

● open access yes/ no should be included as a characteristic and open access and non open access abstracts should be compared

Response: based on reviewer suggestion it was added and highlighted.

● Please define when a CONSORT-A item was reported adequately

Response: A CONSORT-A item was considered reported adequately when the abstract provided clear and complete information in accordance with the specific guideline outlined by CONSORT for abstracts. This entails fulfilling the criteria specified for each item, ensuring a comprehensive and transparent presentation of essential study details within the limitations of the abstract format.

● Is the overall reporting score the sum of items? Please define it as it is the primary endpoint of the study. This should be changed in the section about the primary endpoint as this is used to identify the quality.

Response: Yes, the overall reporting score is derived from the sum of items assessed in accordance with CONSORT-A guidelines. It serves as the primary endpoint of the study, indicating the quality of reporting in RCT abstracts. Based on reviewer sugestion it was added.

● To the 3 point in spins in the results section: how can this be checked when abstracts with multiple primary endpoints are excluded?

Response: We have thoroughly reviewed the spin analysis and have identified a few discrepancies that we would like to address. First, we acknowledge that we omitted the sentence "more than one primary outcome" in the spin evaluation methods section. This was an oversight on our part, and we apologize for any confusion it may have caused. 

 Second, we have re-assessed all of the excluded papers that had multiple primary outcomes. Some papers had multiple primary outcomes, but none of these were used in the spin analysis because the main outcomes had significant results. The titles, primary outcomes, and reasons for excluding these studies are as follows:

Postoperative administration of dienogest plus estradiol valerate versus levonorgestrel-releasing intrauterine device for prevention of pain relapse and disease recurrence in endometriosis patients (Primary outcomes: pain and disease recurrence rate) significant result

Treatment of Endometriosis-Associated Pain with Elagolix, an Oral GnRH Antagonist (Primary outcomes: non menstrual pelvic-pain and dysmenorrhea). significant result

Beneficial effects of oral lactobacillus on pain severity in women suffering from endometriosis: A pilot placebo-controlled randomized clinical trial (Primary outcomes: pain severity using (VAS) scores for dysmenorrhea, dyspareunia and chronic pelvic pain) significant result

Effectiveness of the association micronized N-Palmitoylethanolamine (PEA)-transpolydatin in the treatment of chronic pelvic pain related to endometriosis after laparoscopic assessment: A pilot study (Primary outcomes: pelvic pain, dysmenorrhea and dyspareunia). significant result

Letrozole and norethisterone acetate versus letrozole and triptorelin in the treatment of endometriosis related pain symptoms: a randomized controlled trial (Primary outcomes: pain symptoms( significant result

Maintenance therapy with dienogest following gonadotropin-releasing hormone agonist treatment for endometriosis-associated pelvic pain (Primary outcomes: pelvic pain and irregular uterine bleeding) significant result

Effect of abdominal acupuncture on pain of pelvic cavity in patients with endometriosis (primary outcome: the scores of McGill pain questionaire, level of serum CA125, average value of the radial line of endometrial cyst of ovary and the sum of 3 radial lines of the uterus of patients with adenomyosis as the complication) significant result

Antioxidant supplementation reduces endometriosis-related pelvic pain in humans(primary outcomes: peritoneal fluid biomarkers for oxidation or affect pelvic pain) significant result

Repetitive transcranial magnetic stimulation therapy (Rtms) for endometriosis patients with refractory pelvic chronic pain: A pilot study (primary outcomes: tolerance, pain change and Quality of life) significant result

Does Nomegestrol Acetate plus 17β-Estradiol Oral Contraceptive Improve Endometriosis-Associated Chronic Pelvic Pain in Women? (primary outcomes: pain symptoms, Quality of life, Female Sexual Function Index, and Female Sexual Distress Scale) significant result

Treatment of endometriosis-associated pain with linzagolix, an oral gonadotropin-releasing hormone–antagonist: a randomized clinical trial )primary outcomes: number of responders (R30% reduction in overall pelvic pain) after 12 weeks. Other endpoints included dysmenorrhea, non-menstrual pelvic pain, serum estradiol, amenorrhea, quality of life (QoL) measures, and bone mineral density (BMD)).

● The last sentence in the “Spins in conclusions sections” should be part of the exclusion criteria

Response: We had two groups of eligibility criteria, one for quality assessment and the other for spin assessment. The above sentence is the exclusion for spin and concordance with the previous comment we omitted it. 

● Are all points according to spin of the same importance or is there a weighting possible/ necessary?

Response: It was the same.

Statistical analysis:

● Only the primary endpoint/comparison should have a hypothesis testing? → why are there so many tests planned?

Response: All of the tests were performed based on the primary endpoints 

● In order to find associations and influencing factors on reporting quality a regression model (for example poisson regression) should be performed instead of the testing that is used. The authors should fit to regression models (one for CONSORT-A items and one for spin) so that a better (and adjusted) interpretation of the factors is possible. I ask the authors to add to these regression analyses.

Response: we encountered challenges in conducting regression analysis due to the limited sample size of 47 included articles. The small sample size resulted in a low power, making it difficult to draw statistically significant conclusions from the regression models. In accordance with reviewer’s comment we performed it. The result is as follow:

we utilized a multivariable linear regression with the backward method to assess the factors influencing the Quality of Reporting (QoR). We considered the QoR as the dependent variable and included the word count of the abstract, the indexing information of the journal (such as whether it is indexed in PubMed, Scopus, and Embase), and journal metrics (including impact factor (IF), cite score, and H-index) as the independent factors.As seen in table 3, with increasing one unit in word count and also the cite score, the mean change in QoR is increased (beta for word count=0.017, -value<0.001 and beta for cite score=0.688, p-value<0.001).

Results

● The adherence to CONSORT-A should be described in more detail as it is the primary endpoint of the study

Response: It was done so.

● Please delete testing and add the results of regression modeling

Response: We provided a detailed explanation in our previous comments. we utilized a multivariable linear regression with the backward method to assess the factors influencing the Quality of Reporting (QoR).

Discussion

● When you analyze factors that affect the quality then this should be done with regression models.

Response: Although only 47 articles were included in the analysis, we conducted regression analysis despite the small sample size to comply with the reviewer's suggestion. However, the power of the analysis was compromised due to the limited sample size

● General question: Did you check if the item was reported or was the quality and completeness of the reporting checked as well?

Response: We did not only check if the items were reported but also evaluated the quality and completeness of the reporting using the CONSORT-A extension. The CONSORT-A extension includes 11 related sub-items that assess the clarity, accuracy, and consistency of the reported information.

● none of the abstracts presented precise data as required” what do you mean with this statement?

● 

Response: in the statement "none of the abstracts presented precise data as required," we meant to convey that the abstracts of the included RCTs did not adequately provide specific numerical data as required by the CONSORT-A guidelines. 

● “most items identified that were suboptimal” it was not reported at all or insufficiently

Response: it was added to the result section (as follow) and also it was stated in table2 

Furthermore, evaluating each item of CONSORT-A checklist, in titles of 16 (45.7%) papers, identified the study as randomized. In most of abstracts the eligibility criteria for participants was described. Also, in 30 (85.7%) of abstracts the study design was stated in title or in the method section, even though randomization part is a very important, only about half of the articles reported it generally and about 17.14% of them reported it in detail. The detail of CONSORT items is shown in table 2.“

● why are allocation concealment and blinding the most important points?

Response: These two points were the most important to certify the authenticity of results. to identify the selection bias, it is important to indicate how participants were allocated and randomized to the intervention. Understanding this information allows readers to determine whether the findings of the trial are biased in favor of the new therapy

● Please add the information on the medical field that was investigated in reference 17 and 18

Response: It was added and highlighted.

● Why was information on endpoints extracted from the full texts when abstracts were reviewed?

Response: to differentiate the primary outcome/outcomes from secondary outcomes, somtimes it was needed to extract the endpoints from the fulltext. 

● I could not find the results on IF that are mentioned in the discussion.

Response: It was added.

● The study registration is mentioned as a method to improve quality. But how is ensured that these are complete and correct?

Response: it was out of the scope and objectives of our study. We just reported it in table 2.

Conclusion:

● The study only shows the need for improvement of RCT abstracts but not of RCT reports. Please change the conclusion accordingly. 

Response: It was revised.

The authors should revise the language to improve readability in some parts of the manuscript.

Response: It was done

● The study is merthodologically sound and well presented, however, I have one major concern which is the period of study for article selection. The authors indicate that the articles published until Sept 2021 has been included. It is possible that the authors might have missed some crucial data on the trials published in the past 2 years. Hence I suggest the authors to update the search by including articles published until September 2023 and present the updated results.

Response: Extending our search to include articles published until December 2023 was indeed

---

## [Decision Letter · Decision Letter 1]

26 Feb 2024

PONE-D-23-21849R1The reporting quality and spin of randomized controlled trials of endometriosis pain: Methodological study based on CONSORT extension on abstractsPLOS ONE

Dear Dr. Mirabi,

Thank you for submitting your manuscript to PLOS ONE. After careful consideration, we feel that it has merit but does not fully meet PLOS ONE’s publication criteria as it currently stands. Therefore, we invite you to submit a revised version of the manuscript that addresses the points raised during the review process.

**ACADEMIC EDITOR: **

**Please reply to all reviewers comments **

We look forward to receiving your revised manuscript.

Kind regards,

Ahmed Mohamed Maged, MD

Academic Editor

PLOS ONE

Reviewers' comments:

Reviewer's Responses to Questions

**Comments to the Author**

1. If the authors have adequately addressed your comments raised in a previous round of review and you feel that this manuscript is now acceptable for publication, you may indicate that here to bypass the “Comments to the Author” section, enter your conflict of interest statement in the “Confidential to Editor” section, and submit your "Accept" recommendation.

Reviewer #2: (No Response)

Reviewer #3: All comments have been addressed

Reviewer #4: All comments have been addressed

2. Is the manuscript technically sound, and do the data support the conclusions?

Reviewer #2: Partly

Reviewer #3: Yes

Reviewer #4: Yes

3. Has the statistical analysis been performed appropriately and rigorously? 

Reviewer #2: No

Reviewer #3: Yes

Reviewer #4: Yes

4. Have the authors made all data underlying the findings in their manuscript fully available?

Reviewer #2: Yes

Reviewer #3: Yes

Reviewer #4: Yes

5. Is the manuscript presented in an intelligible fashion and written in standard English?

Reviewer #2: Yes

Reviewer #3: Yes

Reviewer #4: Yes

6. Review Comments to the Author

Reviewer #2: Introduction:

Aim (ii) should be answered using a regression model.

Assessment of reporting quality:

Examples in a supplementary file that show examples for adequately AND insufficient reporting. This would help the reader to understand data collection

Main outcome measure is not adequately defined: It is called quality. Please define it more (as a score with the range of possible values).

Statistical Analysis:

The Analysis should be performed in a correct and adequate way. Currently testing is performed without taking into account the problem of multiple testing. (The level of significance and if it was corrected due to multiple testing must be explained)

Please do not use the word “incidence” when describing figure 3

An insufficient linear regression is performed.

Regression analysis should be the main method to answer the question of factors associated with quality of abstract-reporting.

1) More factors should be included in the analysis (all that are interesting and currently used in hypothesis testing)

Word count, FWCI, year, gynecological journal (yes/ no), open access (yes/no), word count, impact factor, cite score

Please perform regression without variable selection and choose all the variables that are currently analysed with tests

If linear regression is inaapropriate please use logistic regression (with a dichotomous quality endpoint using a cutpoint) or poisson regression (see: DOI: 10.1136/bmjopen-2018-021912 as an example)

If this is not available in SPSS use R instead.

2) Analysis of spin (yes/ no) must be conducted with logistic regression and not with the currently performed testing

Reviewer #3: Dear editor

Thank you for the opportunity to review this manuscript.

All feedback and suggestions have been duly noted by the authors with modifications evident in the latest submission. This reviewer has no additional recommendations of note at this juncture.

Reviewer #4: The authors have satisfactorily addressed comments raised in the previous review by updating the search strategy. The article is methodologically sound and I do not have further comments.

7. PLOS authors have the option to publish the peer review history of their article (what does this mean?). If published, this will include your full peer review and any attached files.

Reviewer #2: No

Reviewer #3: **Yes: **Shabnam ShahAli

Reviewer #4: No

---

## [Author Response · Author response to Decision Letter 1]

1 Mar 2024

Reviewer #2: Introduction:

Aim (ii) should be answered using a regression model.

Response: we performed regression analysis and the result were added.

Assessment of reporting quality:

Examples in a supplementary file that show examples for adequately AND insufficient reporting. This would help the reader to understand data collection

Main outcome measure is not adequately defined: It is called quality. Please define it more (as a score with the range of possible values).

Statistical Analysis:

The Analysis should be performed in a correct and adequate way. Currently testing is performed without taking into account the problem of multiple testing. (The level of significance and if it was corrected due to multiple testing must be explained)

Response: Wherever we compare a parameter in multiple subgroups we adjust the level of significance. For example, when we want to compare QoR in subgroups of IF quartiles we used Bonferroni post hoc test to taking into account the problem of multiple testing. We added it in the method section. 

Please do not use the word “incidence” when describing figure 3

Response: we corrected and changed it to spin intensity.

An insufficient linear regression is performed.

Regression analysis should be the main method to answer the question of factors associated with quality of abstract-reporting.

1) More factors should be included in the analysis (all that are interesting and currently used in hypothesis testing)

Word count, FWCI, year, gynecological journal (yes/ no), open access (yes/no), word count, impact factor, cite score

Please perform regression without variable selection and choose all the variables that are currently analyzed with tests

Response: In regression analysis we need more sample size. “the ratio of the sample size to the number of parameters in a regression model should be at least 20 to 1.” (https://doi.org/10.1016/j.aucc.2012.07.002) so we couldn’t perform multivariate regression analysis. But in accordance to the reviewer’s comment, we performed it and added to the result section.

Assessing the factors influencing the QoR in the abstracts of RCTs is crucial for understanding the potential determinants of reporting quality in scientific literature. we employed a multivariable linear regression model to investigate the impact of various independent variables on the QoR of RCT abstracts focusing on endometriosis pelvic pain. The independent variables considered in our analysis included the word count of the abstract, indexing in databases such as PubMed, Scopus, and ISI, Field-Weighted Citation Impact (FWCI), publication in a gynecological journal, open access status, Impact Factor (IF), Cite Score, and H-index.

2) Analysis of spin (yes/ no) must be conducted with logistic regression and not with the currently performed testing.

Response: As in regression analysis we need at least 20 sample sizes per each independent variable, we couldn’t perform multivariate logistic regression. So, we performed univariable logistic regression. The result is added. 

to evaluate the factors influencing spin in the reporting of research findings, logistic regression analysis was conducted. The independent variables considered in the analysis included the word count of the abstract, indexing status in PubMed (yes/no), indexing status in Scopus (yes/no), indexing status in ISI (yes/no), FWCI, designation as a gynecological journal (yes/no), publication in an open-access journal (yes/no), IF, Cite Score, and H-index. These variables were entered into the univariable linear regression model to assess their individual relationships with the presence of spin. However, the analysis did not reveal any statistically significant relationships between the independent variables and the presence of spin in the abstracts. Additionally, the model encountered convergence issues specifically for the variables related to indexing in PubMed, Scopus, and ISI, indicating potential limitations in the modeling approach for these factors

---

## [Decision Letter · Decision Letter 2]

11 Mar 2024

PONE-D-23-21849R2The reporting quality and spin of randomized controlled trials of endometriosis pain: Methodological study based on CONSORT extension on abstractsPLOS ONE

Dear Dr. Mirabi,

Thank you for submitting your manuscript to PLOS ONE. After careful consideration, we feel that it has merit but does not fully meet PLOS ONE’s publication criteria as it currently stands. Therefore, we invite you to submit a revised version of the manuscript that addresses the points raised during the review process.

**ACADEMIC EDITOR: Please respond to all reviewers comments**

We look forward to receiving your revised manuscript.

Kind regards,

Ahmed Mohamed Maged, MD

Academic Editor

PLOS ONE

Journal Requirements:

Reviewers' comments:

Reviewer's Responses to Questions

**Comments to the Author**

1. If the authors have adequately addressed your comments raised in a previous round of review and you feel that this manuscript is now acceptable for publication, you may indicate that here to bypass the “Comments to the Author” section, enter your conflict of interest statement in the “Confidential to Editor” section, and submit your "Accept" recommendation.

Reviewer #2: (No Response)

2. Is the manuscript technically sound, and do the data support the conclusions?

Reviewer #2: Partly

3. Has the statistical analysis been performed appropriately and rigorously? 

Reviewer #2: N/A

4. Have the authors made all data underlying the findings in their manuscript fully available?

Reviewer #2: Yes

5. Is the manuscript presented in an intelligible fashion and written in standard English?

Reviewer #2: No

6. Review Comments to the Author

Reviewer #2: Incomplete sentence“…Also, the decision maker if a full-text publication is worth buying These factors indicate …”

Logistic Regresion is mentioned in the abstract and the results section. It should also be part of the methods.

Change “Furthermore, we found that most articles were single center (n=30, 63.8%)” to

„Furthermore, we found that most articles described single center (n=30, 63.8%) studies”

Page 19 in the results section words like “interesting” should be avoided

Table 3: please explain in methods what Standardized coeffcients means

Please explain how the Bias-corrected and accelerated (Bca) 95% confidence interval was computed. This is unclear as the explanation is missing

7. PLOS authors have the option to publish the peer review history of their article (what does this mean?). If published, this will include your full peer review and any attached files.

Reviewer #2: No

---

## [Author Response · Author response to Decision Letter 2]

18 Mar 2024

Dear reviewer 2

We have carefully considered and incorporated your feedback into our manuscript for the third time. We hope that the revisions made align with your expectations and are deemed acceptable. Thank you for your continued guidance and support throughout this process.

1-Reviewer #2:” Incomplete sentence “…Also, the decision maker if a full-text publication is worth buying These factors indicate …

-Response: Based on the reviewer's comments, the revised paragraph was added.

These inferences are likely higher among physicians in low- and middle-income countries due to the unavailability of full-text publications. Similarly, patients and the public do not always have access to journal articles. Furthermore, determining whether a full-text publication is worth purchasing is crucial. These factors highlight the importance of reporting quality, clarity, and accuracy during the initial evaluation of RCT abstracts.

2-Logistic Regression is mentioned in the abstract and the results section. It should also be part of the methods.

-Response: Based on the reviewer's comments, it was added to the method section.

3-Change “Furthermore, we found that most articles were single center (n=30, 63.8%)” to

„Furthermore, we found that most articles described single center (n=30, 63.8%) studies”

-Response: Based on the reviewer's suggestion, it was changed and highlighted.

4-Page 19 in the results section words like “interesting” should be avoided.

-Response: Based on the reviewer's comment, the revised paragraph was added and highlighted.

Our findings revealed valuable insights into the relationship between these independent variables and the QoR of abstracts

5-Table 3: please explain in methods what Standardized coeffcients means.

-Response: Based on the reviewer's comment, the revised paragraph was added and highlighted.

Furthermore, to evaluate the strength and direction of the relationships between these independent variables and the QoR, we employed standardized coefficients, also known as Beta coefficients, in our multivariable linear regression model. Beta coefficients, or standardized coefficients, serve as a method to assess the magnitude and orientation of the association among variables that are denoted in disparate units or scales. The process of standardizing these coefficients normalizes the variables to a uniform scale, thereby simplifying the process of comparison

6-Please explain how the Bias-corrected and accelerated (Bca) 95% confidence interval was computed. This is unclear as the explanation is missing.

-Response: BCA method allows for the implementation of three corrections to the standard confidence interval. These corrections account for nonnormality, the bias and the nonconstant standard error of the bootstrap distribution, which is often not symmetric in the case of risk measures

---

## [Decision Letter · Decision Letter 3]

28 Mar 2024

The reporting quality and spin of randomized controlled trials of endometriosis pain: Methodological study based on CONSORT extension on abstracts

PONE-D-23-21849R3

Dear Dr. Mirabi,

We’re pleased to inform you that your manuscript has been judged scientifically suitable for publication and will be formally accepted for publication once it meets all outstanding technical requirements.

Kind regards,

Ahmed Mohamed Maged, MD

Academic Editor

PLOS ONE

Additional Editor Comments (optional):

Reviewers' comments:

Reviewer's Responses to Questions

**Comments to the Author**

1. If the authors have adequately addressed your comments raised in a previous round of review and you feel that this manuscript is now acceptable for publication, you may indicate that here to bypass the “Comments to the Author” section, enter your conflict of interest statement in the “Confidential to Editor” section, and submit your "Accept" recommendation.

Reviewer #2: All comments have been addressed

2. Is the manuscript technically sound, and do the data support the conclusions?

Reviewer #2: Yes

3. Has the statistical analysis been performed appropriately and rigorously? 

Reviewer #2: Yes

4. Have the authors made all data underlying the findings in their manuscript fully available?

Reviewer #2: Yes

5. Is the manuscript presented in an intelligible fashion and written in standard English?

Reviewer #2: Yes

6. Review Comments to the Author

Reviewer #2: (No Response)

7. PLOS authors have the option to publish the peer review history of their article (what does this mean?). If published, this will include your full peer review and any attached files.

Reviewer #2: No
